# Adenovirus 36 Seropositivity Is Related to Inflammation and Imbalance Between Oxidative Stress and Antioxidant Status Regardless of Body Mass Index in Mexican Population

**DOI:** 10.3390/cimb47030166

**Published:** 2025-02-28

**Authors:** Omar Arroyo-Xochihua, Cristian Arbez-Evangelista, Edgar Miranda-Contreras, Yeimy Mar De León-Ramírez, Montserrat Díaz-Edgar, Clara Luz Sampieri, Omar Arroyo-Helguera, María Teresa Álvarez-Bañuelos

**Affiliations:** 1Centro de Investigaciones Biomédicas, Universidad Veracruzana, Av. Luis Castelazo Ayala S/N, Col. Industrial Ánimas, Xalapa 91190, Mexico; zs22000317@estudiantes.uv.mx (O.A.-X.); cristian.arbez@outlook.com (C.A.-E.); 2Laboratorio de Biomedicina y Salud Pública, Instituto de Salud Pública, Universidad Veracruzana, Av. Luís Castelazo Ayala S/N, Col. Industrial Animas, Xalapa 91190, Mexico; edmic1387@gmail.com (E.M.-C.); yeimy_mar@hotmail.com (Y.M.D.L.-R.); omarelind20@gmail.com (M.D.-E.); csampieri@uv.mx (C.L.S.); talvarez@uv.mx (M.T.Á.-B.)

**Keywords:** obesity, human adenovirus-36, lipid profiles, overweight, inflammation, oxidative stress

## Abstract

Background: The etiology of obesity has been associated with genetic and epigenetic factors, hormonal changes, unhealthy lifestyle habits, and infectious agents such as human adenovirus-36 (HAdV-36). Viral infections induce reactive oxygen species, and the imbalance between oxidative stress/antioxidant results in fat accumulation. In the Mexican population, little is known about the frequency of HAdV-36 and its effect on the balance between antioxidants and oxidants, inflammation, and metabolic markers. The purpose of our study was to evaluate the frequency of HAdV-36 seroprevalence and its relation to body mass index (BMI), lipid profiles, glucose levels, inflammation, and levels of antioxidants and oxidative stress in a representative sample. A cross-sectional study was carried out on 112 healthy adults between 18 and 28 years old, who were divided into four groups according to their BMI: underweight (BMI < 18.5); normal weight (BMI 18.5–24.9); overweight (BMI ≥ 25); and obese (BMI ≥ 30). Blood samples were taken to evaluate lipid and glucose profiles, as well as antioxidant and oxidative stress status, using colorimetric techniques. Seropositivity for HAdV-36 and levels of TNF-α, IL-6, and cortisol were determined using an enzyme-linked immunosorbent assay. The HAdV-36 frequency was 15.6% in underweight subjects, 18.7% in the normal-weight subjects, 34.37% in the overweight subjects, and 31.24% in the obese subjects. The subjects who were positive for HAdV-36 seroprevalence had increased levels of IL-6, cortisol, and oxidative stress, independently of BMI. The HAdV-36-positive subjects had reduced LDL-C and HDL-C levels only in the low-weight groups. Glutathione and SOD levels increased in the underweight and normal-weight subjects with positive HAdV-36 seroprevalence, while catalase levels decreased in the normal-weight, overweight, and obese subjects. In conclusion, for the first time, an HAdV-36 seroprevalence in the adult Mexican population is reported which was higher and had a relation with the presence of inflammation, alterations in the lipid profile, and imbalance between oxidative stress and antioxidant status, regardless of BMI. The oxidative stress/antioxidant imbalance could be participating in the stimulation of white adipose tissue deposition.

## 1. Introduction

Obesity is a public health problem that is related to degenerative and metabolic disorders. According to data from the World Health Organization (WHO) [1], there are more than 890 million cases of obesity and more than 2.5 billion overweight people aged 18 years or older. Obesity is a multifactorial condition that involves genetic, socioeconomic, dietary, and lifestyle factors [2], and viral infections have been shown to have played a role in the emergence, persistence, and increase in cases of overweight and obesity [3]. Human adenoviruses 5, 35, and 36 have all been associated with obesity, although the type that has a direct correlation with obesity is human adenovirus-36 (HAdV-36) [4].

There are contrasting epidemiological data about HAdV-36 seroprevalence and obesity; for example, in Turkish participants, the frequency of HAdV-36 was 17.5% in adults with obesity and 4% without obesity [5]. In Denmark, a very low prevalence of HAdV-36 infection was found in the obese and normal-weight adult populations (5.7% vs. 10.0%) [6]. In individuals in the Netherlands and Belgium, no significant association between HAdV-36 and obesity was found, and only 5.5% of HAdV-36-positive participants were associated with obesity [7]. In contrast, the seroprevalences reported in students from the University of Wisconsin and the Bowen Center, Naples, Florida, in the United States (11% in nonobese vs. 30% in obese subjects) was higher [8]; these samples include Hispanic or Latin participants in the sample. It has been reported that the prevalence of obesity is higher in Hispanic or Latin people compared with the general population, because of the sum of genetic and environmental factors, as diets tends to be high in carbohydrates in Hispanic or Latin populations [9]. In the country of Mexico, up to 2012, 26 million Mexican adults were overweight and 22 million were obese [10]. In relation to HAdV-36 seroprevalence and obesity, in Mexico, there is no evidence about the prevalence and effects of HAdV-36 and its association with obesity, metabolic alterations, and the balance of oxidative stress/antioxidants in the adult population.

In animal and in vitro models, HAdV-36 infection induces adipogenesis, decreases triglyceride and cholesterol synthesis, and increases insulin sensitivity, glucose uptake, and lipogenesis [11,12]. In addition, HAdV-36 infection has been related to the expression of transcription factors involved in the accumulation of triglycerides and the differentiation of 3T3-L1 preadipocytes to mature adipocytes through an adipogenesis mechanism, causing a decreased expression of leptin mRNA in 58% of cases [13]. Furthermore, HAdV-36 infection in rats decreases leptin expression, increasing caloric intake, leading to adipogenesis [14]. Moreover, infection with HAdV-36 in preadipocytes causes viral replication, lipid accumulation, and differentiation of preadipocytes into mature adipocytes [15]. Population studies have reported that seropositivity to HAdV-36 is a risk factor for type 2 diabetes [16], and infection with HAdV-36 has been associated with increased expression of glucose transporter genes such as GLUT1/GLUT4, which help in the transport of muscle glucose to fat cells [14]. These observations have also been reported in epidemiological studies which have shown that people with normal weight and obesity present antibodies against HAdV-36 in their blood, and these high levels of antibodies correlate with changes in body weight associated with overweight and obesity, as well as with metabolic changes, mainly in lipid profiles and glucose levels [17,18,19,20,21,22]. Similarly, positive HAdV-36 seroprevalence has also been associated with low levels of cholesterol and triglycerides in both adults [19,21] and the pediatric population [5].

In viral infections, the accumulation of reactive oxygen species causes oxidative stress, which can lead to depletion of the host’s antioxidant defenses and poor antioxidant status; this oxidizing environment is used by viruses to replicate [23]. Oxidative stress/antioxidant imbalance results from excessive fat accumulation, although oxidative stress leads to weight gain by stimulating white adipose tissue deposition and altering food intake; therefore, a vicious circle is established [24], although it is unknown whether HAdV-36 infection affects the antioxidant/oxidant balance in the population with positive seroprevalence of HAdV-36.

The objective of this research was to study the frequency of seroprevalence of HAdV-36 in a sample of Mexican adults aged 18–28 and its relationship with lipid profiles, glucose and insulin levels, inflammation markers, and oxidative stress/antioxidant balance in subjects with low weight, normal weight, overweight, and obesity.

## 2. Materials and Methods

### 2.1. Study Participants

A cross-sectional, exploratory, and analytical study was carried out in 112 adult students from the University of Veracruz, Mexico. Participants were included in this study if they were between 18 and 28 years of age, actual college students from Universidad Veracruzana, and residents of urban areas, with any BMI. The volunteers provided written informed consent to participate and answered questions about their current and past medical history, age, sex, location of residence, being physically active for at least 30 min (sedentarism), and career, which were the variables obtained using a sociodemographic questionnaire. Height and weight were measured without shoes. Height was measured to the nearest 0.1 cm, and weight was measured to the nearest 0.1 kg. Next, the standard formula, weight (kg) divided by height (m^2^), was used to calculate BMI, according to the WHO classification [1]. The waist-to-height ratio (WHR) was then calculated, with WHR = waist circumference/hip. WHR values from 0.4 to 0.49 were classified as healthy (no increased health risk), values from 0.4 to 0.59 as increased risk (increased health risk), and values ≥ 0.6 as high risk. Percent body fat (PBF) was measured using a body composition analyzer. Systolic blood pressure (SBP) and diastolic blood pressure (DBP) were measured on the right arm using an electronic sphygmomanometer after the subjects had rested for 10 min. Finally, blood samples were collected by finger prick with full aseptic precautions; after an overnight fast, morning blood samples were taken from the peripheral veins of the adult participants’ veins between 7 and 10 a.m., and serum was extracted into 6-mL BD Vacutainer Tubes (BD, Becton Drive, Franklin Lakes, NY, USA). The blood was centrifuged within 24 h of collection, and plasma was extracted from the blood and placed in 4-mL BD Vacutainer K2E (EDTA) tubes (Plus Blood Collection Tubes, Becton Drive, Franklin Lakes, NY, USA). Serum samples were then stored at −80 °C in accordance with the CLSI EP28A3c guideline.

The exclusion criteria were if the participants had an acute or chronic disease such as diabetes mellitus type 2, thyroid pathologies, or hypertension; took anti-inflammatory, antidiabetic, or antihypertensive medicaments; consumed alcohol or smoked; were professional athletes; were under dietary therapy; or lived in rural areas. The elimination criteria were that the participants did not give a blood sample or hemolyzed blood sample.

The sample size was calculated by the Raosoft calculator with a confidence level of 95%, keeping a 5% margin error (maximum acceptable), and the minimal sample size was 109 participants. The student participants were recruited using non-probability purposive sampling from the general student list and were contacted at school to schedule a first interview where the inclusion and exclusion criteria were assessed through a data form, including (a) sociodemographic data and (b) clinical antecedents and written informed consent.

This study was approved by the ethics committee of the Psychology department of Universidad Veracruzana, Xalapa region, Veracruz, Mexico, with registration number CEI-FP008/2018, and the investigation committee of the Public Health Institute from Universidad Veracruzana, Xalapa region, Veracruz, Mexico, with registration number 00004097/2018.

### 2.2. IgG Seropositive to HAdV-36 and Inflammation Determination

The quantitative determination of IgG against HAdV-36 in the serum samples was performed using the immune enzymatic assay ELISA kit MyBiosurce Adenovirus 36 (MyBiosurce, San Diego, CA, USA), with positive antibody titers of 1:160 and 1:1250. Inflammatory status was measured by determining interleukin 6 (IL-6) and tumor necrosis factor alpha (TNF-α). For the determination of IL-6, plasma was used, using the ELISA technique according to the manufacturer’s instructions (Merck, México state, Mexico) (Sigma-Aldrich). The change in absorbance was detected at 450 and 550 nm on a Biotek reader. Plasma was used to determine TNF-α, and the manufacturer’s recommendations were followed with the Human Tumor Necrosis Factor α ELISA Kit at an OD of 450 nm (Sigma-Aldrich).

### 2.3. Lipid Profile Determination

Venous blood samples were collected, and total cholesterol levels were determined by colorimetric methods at an OD of 570 nm following the manufacturer’s recommendations; a total Cholesterol Quantification Kit (Sigma-Aldrich) was used for them. The values are reported as mg/dL, as follows: 1, desirable ≤ 200; 2, moderately desirable 200–239; and 3, high ≥ 240. Total triglyceride levels were determined by colorimetric methods at an OD of 540 nm using a biochemical Triglyceride Quantification Kit (Merck, México state, Mexico). Values are reported as mg/dL, as follows: 1, desirable < 150; 2, moderately desirable < 150–199; and 3, high > 200.

### 2.4. Glucose and Insulin Determination

Glucose levels were measured by taking capillary glucose by pricking a finger and taking a drop of blood with a reagent strip that was subsequently read using a glucometer. The values are reported as mg/dL, as follows: 1, desirable ≤ 110; 2, moderately desirable 110–126; and 3, elevated ≥ 126. Insulin levels were measured by colorimetric methods in plasma, with the Merck Human Insulin ELISA Kit, following the manufacturer’s recommendations, performing a curve with an insulin standard ranging from the range of 4.69–300 μU/mL (Merck, México state, Mexico), considering normal to be 5–25 U/mL and insulin resistance 30 U/mL.

### 2.5. Determination of Antioxidant Status

The total antioxidant status (TAS) was analyzed colorimetrically. Briefly, 1.0 mL of FRP solution (25 mL of 300 mmol/L sodium acetate pH 3.6, 2.5 mL of 50 mmol/L potassium ferrocyanide K3Fe (CN) 6, and 2.5 mL of 20 mmol/L FeCl_3_ 6 H_2_O) was added to the serum sample, or H_2_O, or standard with ascorbic acid, and incubated for 3 min at 37 °C. The absorbance against blank H_2_O was read at 593 nm on a microplate reader (Spectramax Plus; Molecular Devices, Sunnyvale, CA, USA). The TAS in the unknown samples was determined by interpolation using a seven-point calibration curve of known amounts of ascorbic acid and expressed in mmol of ascorbic acid equivalents/liter. The enzymatic activity of superoxide dismutase (SOD) was determined by adding 2 mM of pyrogallol to a microplate for superoxide generation, 24 mM of the tetrazolium dye (MTT), 3-(4-5dimethyl thiazol 2-xl), 2,5 diphenyl tetrazolium bromide, and 100 μL of the sample or blank (10 mM TRIS). The reaction was incubated for 10 min and dissolved with 100 μL of DMSO. Absorbance was measured at 570 nm in a microplate reader (Spectramax Plus; Molecular Devices, Sunnyvale, CA, USA). The activity was expressed as U/g of hemoglobin. The determination of catalase activity (CAT) was carried out using 50 μL of plasma, 450 μL of phosphate buffer pH 5.6, and 2.5 mL of 11 mM H_2_O_2_. Kinetics was performed for 1 min at a wavelength of 240 nm in a Genesys TM 10S UV-Visible Spectrophotometer (Thermo-Fisher, San José, CA, USA). The reaction began when H_2_O_2_ was added, and then the absorbance reading began immediately. The reaction blank was prepared for each sample and consisted of 50 μL of hemolyzed human blood, 450 μL of phosphate buffer, and 2.5 mL of injectable water. Finally, the concentration of total glutathione (GSH) was determined using Ellman’s reagent (5,5-dithio-bis-(2-nitrobenzoic acid)). GSH stock solution was prepared for the calibration curve (1.5 mM, 1.25 mM, 1 mM, 0.75 mM, 0.50 mM, and 0.25 mM). The wavelength for reading was 420 nm.

### 2.6. Oxidative Stress and Stress Measurement

The MDA levels, as a lipid peroxidation marker, were measured using 90 μL of plasma extract in 150 mM TRIS buffer with a pH of 7.5, which added to the mixture of 0.4% thio barbituric acid and 20% acetic acid at a pH of 3.0. All samples were heated at 100 °C for 45 min and cooled on ice, and 1% potassium chloride KCl was added. After centrifugation, 180 μL of supernatant was added and measured at 532 nm in a microplate reader (Spectramax Plus; Molecular Devices, Sunnyvale, CA, USA). The results were expressed in absorbance units per 0.1 mL of nmol/mg protein. Basal cortisol in the serum was measured by electrochemiluminescence immunoassay. Basal cortisol in the serum was obtained from blood samples within a time window of 7 am until 10 am. Serum cortisol (µg/dL) was measured using a chemiluminescence immunoassay kit.

### 2.7. Statistical Analysis

The data obtained in this study were analyzed using IBM SPSS version 23.0 software (IBM Corporation, Armonk, NY, USA). Continuous variables were presented as median (minimum–maximum) values, and categorical variables as number (n) and percentage (%). Normal distribution was assessed by the Kolmogorov–Smirnov test. Groups with normal distribution were compared using the Mann–Whitney U-test and Chi-square test to compare categorical variables. The value of *p* < 0.05 was considered statistically significant. The relation between positive or negative HAdV-36 seroprevalence with inflammatory, lipid and glucose profile, oxidative stress and antioxidant status, and BMI was by Spearman’s Rho grading test, crude and adjusted linear regression analysis estimating the effect just of age, sex, Adv36 serology and sedentarism status. For this, adjusted odds ratios were used and 95% confidence intervals (95% CI) were obtained. A value of *p* ≤ 0.05 was shown to be statistically significant.

## 3. Results

### 3.1. Sociodemographic Status

A total of 112 participants were included, with 50% being female and 50% male (this was a random fact). Table 1 shows that the age range was 18 to 28 years, with a median age of 19.53 years. Table 2 presents BMI values, and 10% of the participants were underweight, 28% overweight, and 15% obese. The proportion of participants with a high PBF was 42%, and the proportion with an increased risk to health (WHR) was 48%.

### 3.2. Relation Between Positive Seroprevalence for HAdV-36 and BMI

The data presented in Table 3 show a positive relation between increased weight and positive HAdV-36 seroprevalence (*p* = 0.001).

### 3.3. HAdV-36 Positivity Is Related to Lipid Profile and to Levels of Insulin and Glucose

Lipid profiles and glucose levels were related to the BMI of subjects with and without positive seroprevalence for HAdV-36. Table 4 shows that there were no significant differences in the levels of total cholesterol and triglycerides when comparing the group of subjects with and without seroprevalence for HAdV-36 and BMI. However, subjects with positive HAdV-36 seroprevalence and low weight had low levels of LDL-C and HDL-C. In relation to fasting glucose and fasting insulin levels, no significant differences were found among the groups (Table 4).

### 3.4. Relationship Between WHR, PBF, BMI, and Antibody Titers Against HAdV-36 and Confused Risk Factors

Table 5 shows that the antibody titers of 1:160 and 1:1250 against HAdV-36 were significantly related to ≥25.0 kg/m^2^ BMI and PBF. In contrast, the prevalence was 28.5% BMI (18.8% BMI > 25 vs. 9.7% BMI < 25), with no body fat excess. A logistic regression analysis was conducted, and the results showed a statistically significant correlation between these variables and antibody titers with OR = 3.55 (95% CI 1.5–8.4). Based on the presence of HAdV-36 antibodies, the results indicated that WHR values greater than 0.5 were associated with higher antibody titers than WHR values less than 0.5 (*p* = 0.039, 0.032 vs. control group). A crude and adjusted linear regression analysis was undertaken with age, sex, HadV-36, and sedentarism as risk factor variables for weight excess as BMI. In the crude analysis model, BMI > 25 and age had no statistically significant odds, OR = 1.00 (95% CI: 0.8–1.24; *p* = 0.270), nor when adjusted for age, OR = 0.95 (95% CI: 0.72–1.23; *p* < 0.190). The relation with sex and BMI > 25 did not reach statistical significance, crude OR 0.78 (95% CI 0.6–1.01; *p* = 0.321) and adjusted OR = 0.52–0.96; *p* = 0.321. The relation between sedentarism and BMI > 25 was not statistically significant, OR 1.21 (95% CI 1.00–1.47; *p* = 0.055); however, following adjustment for sedentarism and BMI > 25, this association is significant, OR = 1.68 (95% 1.05–1.56; *p* = 0.033). Finally, in the crude analysis, the relation between positive HAdV-36 and BMI > 25 had statistically significant odds for obesity, OR 1.42 (95% 1.15–1.76; *p* = 0.016), adjusted for BMI > 25, OR = 3.37 (95% 1.11–12.51; *p* = 0.015), following further adjustment for low weight and HAdV-36, OR: −1.21, −1.94–0.48; *p* = 0.001.

Table 6 shows the relationships among the BMI groups with subjects with and without positive HAdV-36 seroprevalence. Specifically, cortisol levels were found to be higher in all the HAdV-36-positive subjects, while individuals with normal weight, overweight, and obesity had significantly higher levels of IL-6, compared to the subjects negative for HAdV-36 (Table 6). The differences between TNF-α values in the HAdV-36-positive and HAdV-36-negative participants were not significant.

The data in Table 7 indicate that the participants with positive HAdV-36 seroprevalence had significantly higher levels of oxidative stress across all BMI categories. The HAdV-36-positive underweight and normal-weight participants had higher levels of total glutathione and SOD activity. The subjects with normal weight, overweight, and obesity who were positive for HAdV-36 showed low levels of catalase activity, compared with the subjects negative for HAdV-36 (Table 7).

Table 8 shows multivariable logistic regression analysis with BMI, underweight, and sedentarism to positive HAdV-36 seroprevalence as dependent variables. IL-6 was directly related to BMI > 25 and sedentarism. Cortisol and oxidative stress were directly related to BMI > 25 and underweight, while, in sedentarism, cortisol was not related. Catalase, SOD, and total glutathione were related to underweight but not sedentarism. Catalase was directly associated with BMI > 25.

## 4. Discussion

For the first time, the HAdV-36 seroprevalence in the Mexican population was described, and we found a high rate of HAdV-36-positive prevalence in a sample of Mexican adults between 18 and 28 years old. These prevalence rates were 15% in underweight people (these data were novel), 18% for normal-weight individuals, and 31% for obese subjects, indicating that obese people had higher HAdV-36 seroprevalence compared with people of normal weight. This data contrasts with the much lower findings previously described in other countries; for example, the 2015 Turkish report found a frequency of 17.5% in adult participants with obesity and 4% without obesity [5]. In our study, 15% of underweight subjects tested positive for HAdV-36. In Denmark, a very low prevalence of HAdV-36 infection was found in the obese and normal-weight adult populations (5.7% vs. 10.0%) [6]. In individuals in Belgium and the Netherlands, no significant associations between HAdV-36 and obesity were found, and only 5.5% of the HAdV-36-positive participants were associated with obesity [7]. In the present study, we demonstrated that the prevalence rate of HAdV-36 in the Mexican population is 28%; this high prevalence could explain part the etiology of obesity in Mexico, a country with the second-highest rate of obesity worldwide [10], and accords with the seroprevalences reported in students from University of Wisconsin and the Bowen Center, Naples, Florida, in the United States (11% in nonobese vs. 30% in obese subjects) [8]; these prevalence similarities could be due to the Hispanic or Latin participants in the sample. Another study with higher HAdV-36 seroprevalence was in the Chilean population, with 58% in the obese group [21]. The obesity prevalence is higher in Hispanic or Latin people compared with the general population, due to a sum of risk factors, such as genetic, for example, single-nucleotide polymorphisms have been associated with obesity, high triglyceride levels, and insulin resistance in Mexican American people [25]; environmental factors, as diets tends to be high in carbohydrates in Hispanic or Latin populations [9]; and infection agents such as HAdV-36.

In this study, we did not find significant changes in glucose and insulin levels in HAdV-36 seroprevalence-positive subjects compared to negative subjects. This contrasts with the findings of a study of obese Chilean subjects with positive HAdV-36 seroprevalence, which was associated with insulin resistance and low glucose and insulin levels [21]. Seroprevalence of HAdV-36 has even been associated as a risk factor for type 2 diabetes mellitus [16]. In vitro studies have shown that HAdV-36 infection induces glucose transport in fat cells [14]. In this study, it was a surprise that no significant differences were found in triglyceride levels in both groups with HAdV-36-positive and HAdV-36-negative seroprevalence because children from Chilpancingo, Guerrero, Mexico, showed a decrease in HDL levels, compared with an HAdV-36-negative group [26]. Interestingly, in our study, we did find that HDL-C and LDL-C levels decreased only in subjects with low-weight BMI and positive HAdV-36 seroprevalence. These data are in accord with the lower concentrations of triglycerides and VLDL cholesterol in lean subjects [21] and data from hamsters infected with HAdV-36; both lipoproteins had a negative association with seroprevalence for HAdV-36 and coincided with a change in the lipid profile [27]. In chickens, monkeys, and mice, the mechanism of infection with HAdV-36 involves adipogenesis, decreases lipolysis, and increased lipogenesis [11,12].

Obesity is associated with inflammatory alterations; in the present study, subjects with positive HAdV-36 seroprevalence, normal weight, overweight, and obesity had high levels of IL-6, while, in subjects with low weight, the IL-6 levels were not significant. These data are consistent with high IL-6 levels in children positive for HAdV-36 [28,29] but, contrasting with HAdV-36-infected Wistar rats, there were no significant changes in IL-6 and TNF-α [30].

The imbalance between oxidative stress and antioxidants could result from excessive fat accumulation, although oxidative stress leads to weight gain by stimulating white adipose tissue deposition and altering food intake [24]. In viral infections, the accumulation of reactive oxygen species causes oxidative stress, which can lead to the depletion of the host’s antioxidant defenses; this oxidizing environment is used by viruses to replicate, as reported for the H5N1 influenza virus [23]. However, it is unknown whether this mechanism of action occupies HAdV-36 for its replication. In the present study, novel results were found in the subjects with positive seroprevalence for HAdV-36; there were higher levels of oxidative stress, elevated values of cortisol, and low levels of catalase activity in the overweight and obese subjects with positive seroprevalence for HAdV-36. In addition, the subjects with low weight and normal weight who were positive for HAdV-36 presented high levels of glutathione and SOD activity. In the literature, there is no evidence about HAdV-36 and oxidative stress, elevated values of cortisol, and low levels of catalase activity in overweight and obese subjects with positive seroprevalence for HAdV-36. In addition, the subjects with low weight and normal weight who were positive for HAdV-36 presented high levels of glutathione and SOD activity. In the literature, no evidence about HAdV-36 and oxidative stress/antioxidant balance was found; however, the imbalance between oxidative stress and antioxidant status results from excessive fat accumulation, and oxidative stress leads to weight gain by stimulating white adipose tissue deposition and altering food intake [24]; therefore, it is possible that HAdV-36 contributes to this established vicious circle.

The present study may limit our findings to participants from rural areas; this was a cross-sectional study and does not include a large random sample, and thus a causality between BMI and HAdV-36 cannot be established. Prospective studies are proposed that enroll the Mexican population to confirm a causal relationship between these variables. In addition to this, the study design may have influenced the relation of the presence of altered BMI with HAdV-36 with dietary patterns not influenced by demographic variables related to a BMI increase. However, for first time, the seroprevalence of HAdV-36 in a sample of adult apparently healthy Mexican students is known. In addition, HAdV-36 seroprevalence had a relation with altered lipid profiles, inflammation, and imbalance between antioxidant and oxidative stress status; this is novel data; these change independently of BMI. This study provides the necessary evidence to initiate new research to understand the mechanism associated between HAdV-36 and oxidative stress and antioxidant status and consider it in future health interventions or prospective studies with Hispanic or Latin subjects.

## 5. Conclusions

The seroprevalence of HAdV-36 in the adult Mexican population is reported for the first time and was higher and is present in all four BMI categories, including underweight. The positive seroprevalence for HAdV-36 had a relation with lipid and inflammation alterations. Finally, it is reported for the first time that there is a relation between oxidative stress and antioxidant imbalance against HAdV-36 seroprevalence.

## 6. Recommendations

With these results, we can emphasize the need to carry out continuous research to develop comprehensive intervention programs for the prevention and treatment of overweight and obesity in Hispanic or Latin populations, which may include appropriate nutrition strategies (antioxidants), physical exercise, and antiviral therapy. Both in vivo and in vitro studies are required to understand the HAdV-36 action mechanism on the balance of oxidative stress and antioxidant status, since HAdv-36 could be using oxidative stress in the deposition of white adipose tissue.

## Figures and Tables

**Table 1 cimb-47-00166-t001:** Sociodemographic data.

	Total (%)
Sex	
Female	56 (50)
Male	56 (50)
Age (Range)	18–28
Mean ± Standard deviation	19.53 ± 2.27

**Table 2 cimb-47-00166-t002:** Anthropometric values from the participants.

Variable	n = 112(%)
Body Max Index (BMI)	
Underweight	12 (10.7)
Norm weight	51 (45.5)
Overweight	32 (28.6)
Obesity	17 (15.2)
Percent Body Fat (PBF)	
Down	7 (6.3)
Normal	57 (50.9)
High	48 (42.8)
Waist-to-Height Ratio (WHR)	
0.4 to 0.49 healthy	58 (51.8)
>0.5 increased risk	54 (48.2)

Abbreviations: BMI, body mass index; PBF, percent body fat; WHR, waist-to-height ratio.

**Table 3 cimb-47-00166-t003:** Relation between sociodemographic, anthropometric, and seroprevalence for HAdV-36.

Variable	HAdV-36 Negative (%) n = 80	HAdV-36 Positive (%) n = 32	*p* Value
Sex			0.209
Female	43 (53.75)	13 (40.63)	
Male	37 (46.25)	19 (59.37)	
BMI			0.001
Underweight	7 (8.75)	5 (15.63)	
Normal weight	45 (56.25)	6 (18.75)	
Overweight	21 (26.25)	11 (34.37)	
Obesity	7 (8.75)	10 (31.25)	

The variables sex, age, and BMI were used to separate the groups according to positive or negative seroprevalence of HAdV-36 subjects. The characteristics of participants with positive vs. negative seroprevalence for HAdV-36 were compared using Chi square.

**Table 4 cimb-47-00166-t004:** Relation between lipid and glucose profiles with HAdV-36 seroprevalence with BMI.

	HAdV-36 Negativity (%) (n = 80)				HAdV-36 Positivity (%) (n = 32)			
	Under Weight n = 7	Normal Weight n = 45	Overweight n = 21	Obesity n = 7	Under Weight n = 5	Normal Weight n = 6	Overweight n = 11	Obesity n = 10
Tg (mg/dL)	91.3 ± 3.8	102.5 ± 6.2	101 ± 2.9	119.1 ± 7.4	89.3 ± 4.2	93.3 ± 5.72	111.3 ± 3.2	126.2 ± 5.6
TC (mg/dL)	128 ± 38.3	178 ± 32.1	151 ± 41.4	191 ± 42.1	138 ± 41.8	167 ± 43.1	157 ± 35.1	195 ± 52.2
HDL-C (mg/dL)	51.3 ± 1.3	53.8 ± 1.5	51.5 ± 1.8	55.5 ± 2.1	39.1 ± 1.1 **	43.6 ± 1.7 **	52.7 ± 2.1	53.4 ± 1.9
LDL-C (mg/dL)	93.3 ± 2.9	104 ± 3.5	105 ± 3.0	106.5 ± 4.1	65.9 ± 2.9 **	68 ± 3.1 **	101.9 ± 2.8	102.3 ± 4.1
F. I (µU/mL)	4.9 ± 0.16	5.2 ± 0.12	5.4 ± 0.21	5.6 ± 0.17	4.92 ± 0.13	5.22 ± 0.37	5.1 ± 0.21	5.5 ± 0.2
F. G (mg/dL)	73.1 ± 1.1	84 ± 1.18	91.3 ± 1.3	93.1 ± 1.7	71.3 ± 1.3	80.5 ± 2.1	83.1 ± 4.1	90.9 ± 2.3

BMI and seroprevalence were used to separate the groups. Tg—Triglycerides; TC—Total cholesterol; F.I—Fasting insulin; F.C—Fasting glucose. Chi square was used to compare HAdV-36 positive subjects versus HAdV-36-negative subjects. ** *p* < 0.01 were considered statistically significant.

**Table 5 cimb-47-00166-t005:** Relationship between anthropometric variables and HAdV-36 antibody titers.

Variable	% HAdV-36 Negativity n = 80	% HAdV-36 ab Titters 1:160 n = 15	% HAdV-36 ab Titters 1:1250 n = 17
BMI			
≤24.9 kg/m^2^	52 (65)	7 (46.7)	4 (23.5)
≥25.0 kg/m^2^	28 (26.25)	8 (53.3) *	13 (17.6) *
Percentage of body fat (PBF)			
No body fat excess	50 (62.5)	10 (66.7)	4 (23.5)
Body fat excess	30 (37.5)	5 (33.3) *	13 (76.5) **
WHR			
≤0.5	46 (57.5)	8 (53.4)	4 (23.5)
≥0.5	34 (42.5)	7 (46.6) *	13 (76.5) **

ab—Antibody; HAdV-36—adenovirus-36; %—percentage. Chi square, * *p* = 0.05; ** *p* = 0.001 vs. negative HAdV-36.

**Table 6 cimb-47-00166-t006:** Relationship between inflammatory markers, BMI, and HAdV-36 seroprevalence.

	BMI	Negative HAdV-36	Positive HAdV-36	*p* Value
		Mean ± SD	Mean ± SD	
IL-6 (pg/mL)	Underweight	0.234 ± 0.05	0.278 ± 0.07	0.336
	Normal weight	0.251 ± 0.08	0.413 ± 0.03	0.001
	Overweight	0.396 ± 0.17	0.493 ± 0.02	0.05
	Obesity	1.138 ± 0.07	1.712 ± 0.07	0.004
TNF-α (pg/mL)	Underweight	32.9 ± 3.02	35.3 ± 2.19	0.632
	Normal weight	36.3 ± 1.05	36.1 ± 2.6	0.895
	Overweight	39.3 ± 1.3	40.8 ± 1.7	0.562
	Obesity	42.9 ± 3.25	44.7 ± 4.13	0.169
Cortisol (μg/dL)	Underweight	19.7 ± 0.3	26.3 ± 3.1	0.05
	Normal weight	15.9 ± 0.6	32.8 ± 2.5	0.001
	Overweight	15.5 ± 0.9	39.9 ± 1.3	0.001
	Obesity	21.9 ± 0.4	41.8 ± 3.8	0.001

Subjects were grouped by BMI and seroprevalence. Chi square was used to compare HAdV-36-positive subjects versus HAdV-36-negative subjects.

**Table 7 cimb-47-00166-t007:** Relationship between oxidative stress, antioxidants, BMI, and HAdV-36.

	BMI	Negative HAdV-36	Positive HAdV-36	*p* Value
		Mean ± SD	Mean ± SD	
Oxidative stress (μm/dL MDA)	Underweight	5.9 ± 0.09	7.13 ± 0.3	0.05
	Normal weight	6.6 ± 0.1	8.9 ± 0.43	0.001
	Overweight	5.91 ± 1.2	8.3 ± 0.29	0.05
	Obesity	7.7 ± 0.26	9.9 ± 0.2	0.05
SOD (U/g Hb)	Underweight	82.3 ± 2.5	96.9 ± 2.4	0.05
	Normal weight	92.8 ± 1.7	99.6 ± 4.6	0.05
	Overweight	83.1 ± 2.3	76.7 ± 2.8	0.894
	Obesity	68.9 ± 3.2	63 ± 1.7	0.257
Catalase CAT (units mg/Hb)	Underweight	4.1 ± 0.3	4.3 ± 1.7	0.133
	Normal weight	3.8 ± 0.1	2.0 ± 0.1	0.001
	Overweight	4.3 ± 0.7	2.7 ± 0.3	0.05
	Obesity	3.7 ± 0.1	2.2 ± 0.13	0.001
Total glutathione (nmol GSH/mL)	Underweight	191.1 ± 6.8	231.5 ± 7.4	0.05
	Normal weight	187.4 ± 5.2	271.6 ± 12.5	0.001
	Overweight	168.5 ± 7.3	181.2 ± 9.19	0.323
	Obesity	260.5 ± 10.8	291.5 ± 10.9	0.223

Subjects were grouped by BMI and seroprevalence. MDA— Malondialdehyde; HAdV-36—Human adenovirus-36; SOD—Superoxide dismutase; CAT— Catalase; GSH—Total glutathione. Chi square was used to compare oxidative stress and antioxidant markers with BMI and HAdV-36-positive and HAdV-36-negative subject groups.

**Table 8 cimb-47-00166-t008:** Multivariate logistic regression analysis with sex, HAdV-36, BMI status, and sedentarism as dependent variables.

	OR	95% (CI) *p*-Value
Dependent variable: BMI ˃ 25; ref: positive HAdV-36		
IL-6 (pg/mL)	0.83	(0.75–1.13) *p* = 0.023
TNF-α (pg/mL)	1.03	(0.96–1.11) *p* = 0.521
Cortisol (μg/dL)	1.09	(1.02–1.16) *p* = 0.001
*Oxidative stress* (μm/dL MDA)	0.99	(0.94–1.01) *p* = 0.001
*SOD* (U/g Hb)	0.75	(0.31–1.99) *p* = 0.570
*Catalase* (units mg/Hb)	1.11	(1.08–1.14) *p* = 0.001
*Total glutathione* (nmol GSH/mL)	0.71	(0.22–1.83) *p* = 0.603
Dependent variable: underweight; ref: positive HAdV-36		
IL-6 (pg/mL)	0.87	(0.41–1.78) *p* = 0.785
TNF-α (pg/mL)	0.88	(0.52–1.92) *p* = 0.061
Cortisol (μg/dL)	1.16	(1.12–1.20) *p* = 0.001
Oxidative stress (μm/dL MDA)	0.87	(0.73–0.99) *p* = 0.011
SOD (U/g Hb)	0.98	(0.94–1.06) *p* = 0.017
*Catalase* (units mg/Hb)	1.02	(1.00–1.04) *p* = 0.086
Total glutathione (nmol GSH/mL)	1.03	(1.01–1.06) *p* = 0.013
Dependent variable: sedentarism; ref: positive HAdV-36		
IL-6 (pg/mL)	0.81	(0.75–0.96) *p* = 0.023
TNF-α (pg/mL)	1.01	(0.97–1.12) *p* = 0.200
Cortisol (μg/dL)	1.73	(0.84–3.59) *p* = 0.125
Oxidative stress (μm/dL MDA)	1.01	(1.01–1.03) *p* = 0.022
SOD (U/g Hb)	0.97	(0.91–1.08) *p* = 0.603
Catalase (units mg/Hb)	0.98	(0.92–1.08) *p* = 0.734
Total glutathione (nmol GSH/mL)	0.53	(0.05–4-61) *p* = 0.613

OR, odds ratio; CI, confidence interval. Multivariate regression analyses to BMI > 25, underweight, and sedentarism.

## Data Availability

All the data are contained within this article.

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
