# Peer review of "Adenovirus 36 Seropositivity Is Related to Inflammation and Imbalance Between Oxidative Stress and Antioxidant Status Regardless of Body Mass Index in Mexican Population"

_cimb, 2025, doi:10.3390/cimb47030166_

Round 1

Reviewer 1 Report

Comments and Suggestions for Authors

The article addresses an important topic and is generally interesting.

However, I have a few reservations.

There are many articles on the relationship between Adenovirus 36 prevalence and association with human obesity - if so, what is the novelty of this work? Do the obtained results differ concerning other nationalities?

The title is definitely too long and lacks information that these are studies in the Mexican group. In addition, as the researchers themselves noticed, they studied correlations that do not indicate cause-effect relationships - the title in this context is too unambiguous.

My next question concerns recruitment. What did recruitment to the study look like, why were the age criteria of 18-28 (?) used, who was excluded, and how did it happen that it was possible to accurately select 50% of women and men? Was the existence of additional diseases analyzed? I mean diabetes, hypertension, chronic inflammation - on this topic - there is no information here. Were the people taking medications that could affect the studied lipid metabolism, oxidative stress, and inflammation?

In Table 4 - the use of the notation HDL and LDL is incorrect - they should be changed to HDL-C and LDL-C (because the authors did not examine the concentration of the entire molecule, but cholesterol in lipoprotein).

I suggest that in the discussion the researchers refer to other populations, then it can be written in the conclusions that the seroprevalence of HadV-36 is high. Otherwise, it does not appeal to me.
Many such articles could be written examining different groups from different countries, but what is the innovation in this? The authors do not try to understand and delve into the analyzed relationships; the relationships they have detected are rather widely known.

Author Response

Dear Editor and reviewer´s

Thank you very much for taking the time to review this manuscript. Please find the detailed responses below and the corresponding corrections highlighted in red in the re-submitted file:

Author comments and observations:

The article addresses an important topic and is generally interesting.

However, I have a few reservations.

Observation 1).- There are many articles on the relationship between Adenovirus 36 prevalence and association with human obesity - if so, what is the novelty of this work? Do the obtained results differ concerning other nationalities?

Answer: thanks for your comments and observations, the novelties of this work are:

1.- First, this study is done in Mexican population between 18-28 years old adult apparently healthy (is the first study in adult population), and the novelty is that the prevalence rate of positive seroprevalence to HAdV-36 in the Mexican population was 28%, higher than other countries as described in discussion section (Mexico is the second place in obesity in the world) and HAdV-36 seroprevalence could be associated. Also, we found a high rate of prevalence positive to HAdV-36 in 15% in underweight participants, this data was novelty, and like the prevalence reported in a USA study, because the Hispanic or Latin population (Add in discussion section).

2.- Secondly, this study described for the first time statistical relation between underweight population, with positive seroprevalence, lipid levels, inflammation and oxidative stress/antioxidant imbalance (no causality because cross-sectional design has limitations). These oxidative stress/antioxidant imbalances can become chronic stress and be the beginning of many pathologies, in the future they may be the target of health interventions, based on this evidence.

3.- Third, this study described for the first time that seroprevalence is associated with an imbalance in oxidative stress and antioxidant status independently of BMI, other studies describe dyslipidemia or inflammation only.

Observation 2).- The title is definitely too long and lacks information that these are studies in the Mexican group. In addition, as the researchers themselves noticed, they studied correlations that do not indicate cause-effect relationships - the title in this context is too unambiguous.

Answer: Thanks for the observation.

In the discussion section, the limitations of cross-sectional design are mentioned, and the conclusions are limited to this cross-sectional design; however, statistically we describe the strength of association between the dependent and independent variables, which results from the Pearson or Spearman analysis according to the distribution of the sample. For the same reason, the title refers to such associations and not to causality, although to avoid confusion, the word relation will be used in the title and all manuscript, which would be more in line with the strength of association that was statistically calculated in this study.

 In relation to the length of the title, which is long, the authors propose the following titles: Adenovirus 36 seropositivity is related to inflammation, and imbalance between oxidative stress and antioxidant status regardless of body mass index in Mexican population

Observation 3.- My next question concerns recruitment. What did recruitment to the study look like, why were the age criteria of 18-28 (?) used, who was excluded, and how did it happen that it was possible to accurately select 50% of women and men? Was the existence of additional diseases analyzed? I mean diabetes, hypertension, chronic inflammation - on this topic - there is no information here. Were the people taking medications that could affect the studied lipid metabolism, oxidative stress, and inflammation?

Answer: Thank you very much for the observation, the criteria for inclusion and exclusion of the participants were attached to the methodological section, as follows:

A cross-sectional, exploratory, and analytical study was carried out in 112 adults’ students from the University of Veracruz, México. Participants were included in the study if they were between 18 and 28 years of age, be actual college students from Universidad Veracruzana, residents of urban areas with any BMI. The volunteers provided written informed consent to participate and answered questions about their current and past medical history, age, sex, location of residence, be physically active for at least 30 minutes (sedentarism), and career were the variables obtained using a sociodemographic questionnaire. Height and weight were measured without shoes. Height was measured to the nearest 0.1 cm, and weight was measured to the nearest 0.1 kg. Next, the standard formula, weight (kg) divided by height (m2), was used to calculate BMI, according to the WHO classification [1]. The waist-to-height ratio (WHR) was then calculated, with WHR= waist circumference / hip. WHR values from 0.4 to 0.49 were classified as healthy (no increased health risk), values from 0.4 to 0.59 as increased risk (increased health risk), and values ≥ 0.6 as high risk. Percent body fat (PBF) was measured using a body composition analyzer. Systolic blood pressure (SBP), and diastolic blood pressure (DBP) were measured on the right arm using an electronic sphygmomanometer after subjects had rested for 10 min. Finally, blood samples were collected by finger prick with full aseptic precautions for that, after an overnight fast, morning blood samples were taken from peripheral veins of adult participants veins between 7 and 10 a.m., and serum was extracted into 6-ml BD Vacutainer Tubes (BD, Becton Drive, Franklin Lakes, NY, USA). The blood was centrifuged within 24 hours of collection, and plasma was extracted from the blood and placed in 4-ml BD Vacutainer K2E (EDTA) tubes (Plus Blood Collection Tubes, Becton Drive, Franklin Lakes, NY, USA). Serum samples were then stored at −80°C in accordance with the CLSI EP28A3c guideline [25].

The exclusion criteria were if the participants had an acute or chronic disease such as diabetes mellitus type 2, thyroid pathologies, hypertension; took anti-inflammatory, antidiabetic, or antihypertensive medicaments, alcohol or smoked consumption, professional athletes, or under dietary therapy, and live in rural areas. The elimination criteria were that the participants were not giving the blood sample or hemolyzed blood sample.

The sample size was calculated by the Raosoft calculator with a confidence level of 95%, keeping 5% margin error (maximum acceptable), and the minimal sample size was 109 participants. The student participants were recruited using non-probability purposive sampling from the general student list and contacted at school to schedule a first interview where the inclusion and exclusion criteria were assessed through a data form, including: a) sociodemographic data and b) clinical antecedents and written informed consent.

The study was approved by the ethics committee of the Psychology department of Universidad Veracruzana, Xalapa region, Veracruz, México, with registration number CEI-FP008/2018, and the investigation committee of the Public Health Institute from Universidad Veracruzana, Xalapa region, Veracruz, Mexico, with registration number 00004097/2018.

Observations 4: In Table 4 - the use of the notation HDL and LDL is incorrect - they should be changed to HDL-C and LDL-C (because the authors did not examine the concentration of the entire molecule, but cholesterol in lipoprotein).

Answer: thanks for the correction; HDL and LDL annotation were corrected in the manuscript.

Observation 5: I suggest that in the discussion the researchers refer to other populations, then it can be written in the conclusions that the seroprevalence of HadV-36 is high. Otherwise, it does not appeal to me.

Many such articles could be written examining different groups from different countries, but what is the innovation in this? The authors do not try to understand and delve into the analyzed relationships; the relationships they have detected are rather widely known.

Answer: Thanks for the observation

We add this paragraph in the discussion section:

For firs time was described in Mexican population the HAdV-36 seroprevalence and we found a high rate of prevalence positive to HAdV-36 in a sample of Mexican adults between 18 and 28 years old. These prevalence rates were 15% in underweight people (this data was novelty), 18% for normal-weight individuals, and 31% for obese subjects indicated that obese people had higher HAdV-36 seroprevalence, compared with people of normal weight. This data contrasts with the much lower findings than previously described in other countries, for example, the 2015 report Turkish participants a frequency of 17.5% of adults with obesity and 4% without obesity [5]. In our study, 15% of underweight subjects tested positive for HAdV-36 (no obesity group). In Denmark, a very low prevalence of HAdV-36 infection was found in the obese and normal-weight adult pop-ulations (5.7% vs. 10.0%) [6]. In Belgian and Netherland individuals there are not a significant association between HAdV-36, and obesity were found, and only 5.5% HAdV-36 positive participants were associated with obesity [7]. In the present study, we demonstrated that the prevalence rate of HAdV-36 in the Mexican population is 28%, these high prevalence which could explain part the etiology of obesity in Mexico, a country with the second-highest rate of obesity worldwide [10], according with the seroprevalences reported in students from University of Wisconsin and the Bowen Center, Naples, Florida from the United States (11% in nonobese vs 30% in obese subjects) [8], these prevalence similarities could be by the Hispanic or Latin participants in the sample. Another study with higher HAdV-36 seroprevalence was in Chilean population with 58% in obese group [21]. The obesity prevalence is higher in Hispanic or Latin people compared with the general population, by the summary of risk factors as genetic, for example, single‐nucleotide polymorphisms have been associated with obesity, high triglyceride levels, and insulin resistance in Mexican American people [25]. The environment factor as diets tends to be high in carbohydrates in Hispanic or Latin population [9], and the infection agents as HAdV-36.

In this study, we did not find significant changes in glucose and insulin levels in HAdV-36 seroprevalence-positive subjects compared to negative subjects. This contrasts with the findings of a study of obese Chilean subjects with positive HAdV-36 seroprevalence, which was associated with insulin resistance and low glucose and insulin levels [21]. Seroprevalence of HAdV-36 has even been associated as a risk factor for type 2 diabetes mellitus [16]. In vitro studies have shown that HAdV-36 infection induces glucose transport in fat cells [14]. In this study, it was a surprise that no significant differences were found in triglyceride levels in both groups with HAdV-36-positive and HAdV-36-negative seroprevalence. Because in children from Chilpancingo, Guerrero, Mexico showed a decrease in HDL levels, compared with an HAdV-36 negative group [26]. Interestingly, in our study we did find that HDL-C and LDL-C levels decreased only in subjects with low-weight BMI and positive HAdV-36 seroprevalence. These data in accord with the lower concentrations of tri-glycerides and VLDL cholesterol in lean subjects [21] and data from hamsters infected with HAdV-36, both lipoproteins had a negative association with seroprevalence for HAdV-36, and coincides with a change in the lipid profile [27]. In chickens, monkeys and mice, infection with HAdV-36 the mechanism involves adipogenesis, decrease lipolysis, and increase lipogenesis [11, 12].

Obesity is associated with inflammatory alterations, in the present study, subjects with positive HAdV-36 seroprevalence, normal weight, overweight, and obesity had high levels of IL-6, while subjects with low weight the IL-6 levels were not significant. These data are consistent with high IL-6 levels in children positive for HAdV-36 [28, 29] but contrasting in HAdV-36 infected Wistar rats there were no significant changes in IL-6 and TNF-α [30].

The imbalance between oxidative stress and antioxidant could result from excessive fat accumulation, although oxidative stress leads to weight gain by stimulating white adipose tissue deposition and altering food intake [24]. In viral infections, the accumulation of reactive oxygen species causes oxidative stress, which can lead to depletion of the host's antioxidant defenses; this oxidizing environment is used by viruses to replicate [23], as reported for the H5N1 influenza virus [23]. However, it is unknown if this mechanism of action occupies HAdV-36 for its replication. In the present study, novelty results were found in the subjects with positive seroprevalence for HAdV-36 there were higher levels of oxidative stress, elevated values of cortisol and low levels of catalase activity in overweight and obese subjects with positive seroprevalence for HAdV-36. In addition, subjects with low weight and normal weight who were positive for HAdV-36 presented high levels of glutathione and SOD activity. In the literature there is no evidence about HAdV-36 and oxidative stress, elevated values of cortisol and low levels of catalase activity in overweight and obese subjects with positive seroprevalence for HAdV-36. In addition, subjects with low weight and normal weight who were positive for HAdV-36 presented high levels of glutathione and SOD activity. In the literature no evidence about HAdV-36 and oxidative stress/antioxidant balance was found; however, the imbalance between oxidative stress and antioxidant status results from excessive fat accumulation and oxidative stress leads to weight gain by stimulating white adipose tissue deposition and altering food intake [24], therefore it is possible that HAdV-36 contributes to this established vicious circle.

The present study may limit our findings to participants from rural areas, this was a cross-sectional study, and thus a causality between BMI and HAdV-36 cannot be established and does not include a large random sample. It is proposed prospective studies that enrolling Mexican population to confirm a causal relationship between these variables. In addition to this study design may have influenced the relation of the presence of altered BMI with HAdV-36 with dietary patterns and not influenced by demographic variables related to a BMI increase. However, for first time it is know the seroprevalence of HAdV-36 in a sample of adult apparently healthy Mexican students. In addition, HAdV-36 seroprevalence had relation with altered lipid profiles, inflammation and im-balance between antioxidant and oxidative stress status, this one a novelty data, these changes in-dependently of BMI. This study provides the necessary evidence to initiate new research to un-derstand the mechanism associated between HAdV-36 and oxidative stress and antioxidant status and consider it in future health interventions or prospective studies with Hispanic or Latin subjects.

Regards

PhD. Omar Elind Arroyo Helguera

Reviewer 2 Report

Comments and Suggestions for Authors
  1. Clarity and Coherence: The manuscript presents compelling findings, but the logical flow between sections needs improvement. Transitions between Introduction, Methods, Results, and Discussion should be more seamless. The connection between Adenovirus-36 seropositivity, metabolic changes, and oxidative stress should be discussed more concisely and with clearer emphasis.
  2. Introduction & Research Gap: While the Introduction provides adequate background, it lacks a strong justification for the study. Clearly define how this research builds upon previous findings and why it is necessary for advancing knowledge on HAdV-36 and metabolic health.
  3. Methodology Concerns: The description of sample selection is vague. How were participants recruited? What were the exclusion criteria? The biochemical assay procedures should be elaborated upon to ensure reproducibility, particularly regarding inflammatory markers, oxidative stress markers, and lipid profiles. Were any measures taken to control for confounding variables such as diet, lifestyle, or other infections?
  4. Data Presentation & Statistical Analysis: Some figures and tables lack detailed explanations. Axis labels and legends should be clearer. Statistical significance is reported, but confidence intervals and effect sizes are missing in key analyses. Address the potential impact of confounding factors that might influence the results.
  5. Discussion & Interpretation of Results: Some results are overstated without sufficient supporting evidence. Avoid speculation beyond what the data can substantiate. Findings should be more explicitly compared with previous studies. Address discrepancies and highlight novel contributions. Provide a clearer explanation of the clinical or biological implications of the observed associations.
  6. Conclusion & Future Directions: The Conclusion should better summarize key findings and emphasize their broader implications. Future research directions should be expanded, particularly on mechanistic insights into HAdV-36’s role in obesity and metabolic changes.
  7. Language & Formatting Issues: The manuscript contains grammatical and typographical errors that require careful proofreading. Ensure that all references are consistently formatted and correctly cited.
Comments on the Quality of English Language

The manuscript contains grammatical and typographical errors that require careful proofreading. 

Author Response

Dear Editor and reviewer´s

Thank you very much for taking the time to review this manuscript. Please find the detailed responses below and the corresponding corrections highlighted in red in the re-submitted file:

Author comments and observations:

Clarity and Coherence: The manuscript presents compelling findings, but the logical flow between sections needs improvement. Transitions between Introduction, Methods, Results, and Discussion should be more seamless. The connection between Adenovirus-36 seropositivity, metabolic changes, and oxidative stress should be discussed more concisely and with clearer emphasis.

Thank you for taking the time in such a detailed review and excellent observations to improve the manuscript.

Observation: Introduction & Research Gap: While the Introduction provides adequate background, it lacks a strong justification for the study. Clearly define how this research builds upon previous findings and why it is necessary for advancing knowledge on HAdV-36 and metabolic health.

Answer: both in the abstract and in the introduction (text in red) a greater justification has been added, as follows:

Simple Summary:

Human adenovirus-36 is a risk factor for obesity and is associated with inflammation and changes in lipid and glucose profiles. México country occupied the second place in obesity in the world, and it is unknow the HAdV-36 seroprevalence in Mexican adults and its relationship with stress and antioxidant balance. Therefore, we examined in a sample of adult Mexican the relationships between HAdV-36 seroprevalence, body mass index, lipid and glucose profiles, inflammation, and the antioxidants and oxidative stress status.

Introduction section:

They are contrasts epidemiological data about HAdV-36 seroprevalence and obesity, for example, in Turkish participants the frequency of HAdV-36 was 17.5% in adults with obesity and 4% without obesity [5]. In Denmark, a very low prevalence of HAdV-36 infection was found in the obese and normal-weight adult populations (5.7% vs. 10.0%) [6]. In Netherland and Belgium individuals no significant association between HAdV-36 and obesity were found, and only 5.5% HAdV-36 positive participants were associated with obesity [7]. In contrast the seroprevalences reported in students from University of Wisconsin and the Bowen Center, Naples, Florida from the United States (11% in nonobese vs 30% in obese subjects) was higher [8], these sample include Hispanic or Latin participants in the sample. It has reported that the prevalence of obesity is higher in Hispanic or Latin people compared with the general population, because of the summary of genetic, the environment factor as diets tends to be high in carbohydrates in Hispanic or Latin population [9]. In México country, until 2012, 26 million Mexican adults have overweight and 22 million obesity [10]. Related to HAdV-36 seroprevalence and obesity, in Mexico, they are not evidence about the prevalence and effects of HAdV-36 and its association with obesity, metabolic alterations, and in the balance of oxidative stress/antioxidant in adult population.

In viral infections, the accumulation of reactive oxygen species causes oxidative stress, which can lead to depletion of the host's antioxidant defenses and poor antioxidant status; this oxidizing environment is used by viruses to replicate [23]. Oxidative stress/antioxidant imbalance results from excessive fat accumulation, although oxidative stress leads to weight gain by stimulating white adipose tissue deposition and altering food intake; therefore, a vicious circle is established [24], although it is unknown whether HAdV-36 infection affects the antioxidant/oxidant balance in the population with positive seroprevalence to HAdV-36. 

The objective of this research was to study the frequency of seroprevalence for HAdV-36 in a sample of Mexican adults aged 18-28, and its relationship with lipid profiles, glucose and insulin levels, inflammation markers, oxidative stress/antioxidant balance in subjects with low weight, normal weight, overweight and obesity.

Observations: Methodology Concerns: The description of sample selection is vague. How were participants recruited? What were the exclusion criteria? The biochemical assay procedures should be elaborated upon to ensure reproducibility, particularly regarding inflammatory markers, oxidative stress markers, and lipid profiles.

Answer: Thank you for the observations, in the methodology section the required information was included in red, as follows:

A cross-sectional, exploratory, and analytical study was carried out in 112 adults’ students from the University of Veracruz, México. Participants were included in the study if they were between 18 and 28 years of age, be actual college students from Universidad Veracruzana, residents of urban areas with any BMI. The volunteers provided written informed consent to participate and answered questions about their current and past medical history, age, sex, location of residence, be physically active for at least 30 minutes (sedentarism), and career were the variables obtained using a sociodemographic questionnaire. Height and weight were measured without shoes. Height was measured to the nearest 0.1 cm, and weight was measured to the nearest 0.1 kg. Next, the standard formula, weight (kg) divided by height (m2), was used to calculate BMI, according to the WHO classification [1]. The waist-to-height ratio (WHR) was then calculated, with WHR= waist circumference / hip. WHR values from 0.4 to 0.49 were classified as healthy (no increased health risk), values from 0.4 to 0.59 as increased risk (increased health risk), and values ≥ 0.6 as high risk. Percent body fat (PBF) was measured using a body composition analyzer. Systolic blood pressure (SBP), and diastolic blood pressure (DBP) were measured on the right arm using an electronic sphygmomanometer after subjects had rested for 10 min. Finally, blood samples were collected by finger prick with full aseptic precautions for that, after an overnight fast, morning blood samples were taken from peripheral veins of adult participants veins between 7 and 10 a.m., and serum was extracted into 6-ml BD Vacutainer Tubes (BD, Becton Drive, Franklin Lakes, NY, USA). The blood was centrifuged within 24 hours of collection, and plasma was extracted from the blood and placed in 4-ml BD Vacutainer K2E (EDTA) tubes (Plus Blood Collection Tubes, Becton Drive, Franklin Lakes, NY, USA). Serum samples were then stored at −80 °C in accordance with the CLSI EP28A3c guideline.

The exclusion criteria were if the participants had an acute or chronic disease such as diabetes mellitus type 2, thyroid pathologies, hypertension; took anti-inflammatory, antidiabetic, or antihypertensive medicaments, alcohol or smoked consumption, professional athletes, or under dietary therapy, and live in rural areas. The elimination criteria were that the participants were not giving the blood sample or hemolyzed blood sample.

The sample size was calculated by the Raosoft calculator with a confidence level of 95%, keeping 5% margin error (maximum acceptable), and the minimal sample size was 109 participants. The student participants were recruited using non-probability purposive sampling from the general student list and contacted at school to schedule a first interview where the inclusion and exclusion criteria were assessed through a data form, including: a) sociodemographic data and b) clinical antecedents and written informed consent.

The study was approved by the ethics committee of the Psychology department of Universidad Veracruzana, Xalapa region, Veracruz, México, with registration number CEI-FP008/2018, and the investigation committee of the Public Health Institute from Universidad Veracruzana, Xalapa region, Veracruz, Mexico, with registration number 00004097/2018.

Lipid profile determination

Venous blood samples were collected, and total cholesterol levels were determined by colorimetric methods at an OD of 570 nm following the manufacturer's recommendations; total Cholesterol Quantification Kit (Sigma-Aldrich) was used for them. The values are reported as mg/dl, considering: 1, desirable ≤ 200; 2, moderately desirable 200-239; and 3, high ≥240. Total triglyceride levels were determined by colorimetric methods at an OD of 540 nm using a biochemical Triglyceride Quantification Kit (Sigma-Aldrich). Values are reported as mg/dl, being: 1, desirable < 150; 2, moderately desirable < 150-199; and 3, high >200.

2.5. oxidative stress and stress measurement

The MDA levels, as a lipid peroxidation marker were measured using 90 μL of plasma extract in 150 mM TRIS buffer with a pH of 7.5, which added to the mixture of 0.4% thio barbituric acid and 20% acetic acid at pH of 3.0. All samples were heated at 100 °C for 45 min and cooled on ice, and 1% potassium chloride KCl was added. After centrifugation, 180 μL of supernatant was added and measured at 532 nm in a microplate reader (Spectramax Plus; Molecular Devices, Sunnyvale, CA, USA). The results were expressed in absorbance units per 0.1 mL of nmol/mg protein. Basal cortisol in the serum was measured by electrochemiluminescence immunoassay. Basal cortisol in the serum was obtained from blood samples within a time window of 7 am until 10 am. Serum cortisol (µg/dl) was measured using chemiluminescence immunoassay kit.

Observation: Were any measures taken to control for confounding variables such as diet, lifestyle, or other infections?

In the methodology section was add the selection criteria information because in this sutyd was discarded the follow variables: previous chronic diseases (diabetes mellitus type 2, thyroid pathologies, hypertension; took anti-inflammatory, antidiabetic, or antihypertensive medicaments), participants that living in rural areas, consumption of alcohol or smoked, professional athletes, or under dietary therapy (these variables have been reported that modified the BMI).

Methodology section as follow:

A cross-sectional, exploratory, and analytical study was carried out in 112 adults’ students from the University of Veracruz, México. Participants were included in the study if they were between 18 and 28 years of age, be actual college students from Universidad Veracruzana, residents of urban areas with any BMI. The volunteers provided written informed consent to participate and answered questions about their current and past medical history, age, sex, location of residence, be physically active for at least 30 minutes (sedentarism), and career were the variables obtained using a sociodemographic questionnaire. Height and weight were measured without shoes. Height was measured to the nearest 0.1 cm, and weight was measured to the nearest 0.1 kg. Next, the standard formula, weight (kg) divided by height (m2), was used to calculate BMI, according to the WHO classification [1]. The waist-to-height ratio (WHR) was then calculated, with WHR= waist circumference / hip. WHR values from 0.4 to 0.49 were classified as healthy (no increased health risk), values from 0.4 to 0.59 as increased risk (increased health risk), and values ≥ 0.6 as high risk. Percent body fat (PBF) was measured using a body composition analyzer. Systolic blood pressure (SBP), and diastolic blood pressure (DBP) were measured on the right arm using an electronic sphygmomanometer after subjects had rested for 10 min. Finally, blood samples were collected by finger prick with full aseptic precautions for that, after an overnight fast, morning blood samples were taken from peripheral veins of adult participants veins between 7 and 10 a.m., and serum was extracted into 6-ml BD Vacutainer Tubes (BD, Becton Drive, Franklin Lakes, NY, USA). The blood was centrifuged within 24 hours of collection, and plasma was extracted from the blood and placed in 4-ml BD Vacutainer K2E (EDTA) tubes (Plus Blood Collection Tubes, Becton Drive, Franklin Lakes, NY, USA). Serum samples were then stored at −80 °C in accordance with the CLSI EP28A3c guideline.

The exclusion criteria were if the participants had an acute or chronic disease such as diabetes mellitus type 2, thyroid pathologies, hypertension; took anti-inflammatory, antidiabetic, or antihypertensive medicaments, alcohol or smoked consumption, professional athletes, or under dietary therapy, and live in rural areas. The elimination criteria were that the participants were not giving the blood sample or hemolyzed blood sample.

The sample size was calculated by the Raosoft calculator with a confidence level of 95%, keeping 5% margin error (maximum acceptable), and the minimal sample size was 109 participants. The student participants were recruited using non-probability purposive sampling from the general student list and contacted at school to schedule a first interview where the inclusion and exclusion criteria were assessed through a data form, including: a) sociodemographic data and b) clinical antecedents and written informed consent.

The study was approved by the ethics committee of the Psychology department of Universidad Veracruzana, Xalapa region, Veracruz, México, with registration number CEI-FP008/2018, and the investigation committee of the Public Health Institute from Universidad Veracruzana, Xalapa region, Veracruz, Mexico, with registration number 00004097/2018.

In the methodology section in the statistics part, information on how the statistical analysis was carried out was added for the confounding factors as follows:

2.6. Statistical Analysis

Data obtained in the study were analyzed using IBM SPSS version 23.0 software (IBM Corporation, Armonk, NY, USA). Continuous variables were presented as median (minimum-maximum) values and categorical variables as number (n) and percentage (%). Normal distribution was assessed by the Kolmogorov-Smirnov test. Groups with normal distribution were compared using the Mann Whitney U-test and Chi-square test to compare categorical variables. The value of p< 0.05 was considered statistically significant. The relation between positive or negative HAdV-36 seroprevalence with inflammatory, lipid and glucose profile, oxidative stress, and antioxidant status, and BMI was by Spearman's Rho grading test, crude and adjusted linear regression analysis estimating the effect just of age, sex, Adv36 serology and sedentarism status. For this was used and adjusted odds ratios and 95% confidence intervals (95% CI) were obtained. A p-value <0.05 was considered significant. A value of p≤0.05 was shown to be statistically significant.

Observation: Data Presentation & Statistical Analysis: Some figures and tables lack detailed explanations. Axis labels and legends should be clearer. Statistical significance is reported, but confidence intervals and effect sizes are missing in key analyses. Address the potential impact of confounding factors that might influence the results.

Answer: The information was added in the tables where appropriate and in the text was add the impact of confounding factors as follow:

3.4. Relationship between WHR, PBF, BMI, and antibody titers against HAdV-36 and confused risk factors

Table 5 shows that the antibody titers of 1:160 and 1:1250 against HAdV-36 were significantly related to 25.0 kg/m2 BMI and PBF. In contrast, the prevalence was 28.5% BMI (18.8% BMI ˃ 25 vs. 9.7% BMI ˂ 25), with no body fat excess. A logistic regression analysis was conducted, and the results showed a statistically significant correlation between these variables and antibody titers with OR = 3.55 (95% CI 1.5-8.4). Based on the presence of HAdV-36 antibodies, the results indicated that WHR values greater than 0.5 were associated with higher antibody titers than WHR values less than 0.5 (p = 0.039, 0.032 vs. control group). A crude and adjusted linear regression analysis were led with age, sex, HadV-36, and sedentarism as risk factor variables for weight excess as BMI. In the crude analysis model, BMI ˃ 25 and age had no statistically significant odds, OR= 1.00 (95% CI: 0.8-1.24; p=0.270), neither adjusted OR= 0.95 (95% CI: 0.72-1.23; p<0.190) for age. The relation with sex and BMI ˃ 25, did not reach statistical significance, crude OR 0.78 (95% CI 0.6-1.01; p=0.321) and adjusted OR=0.52-0.96; p=0.321). The relation between sedentarism and BMI ˃ 25 was not statistically significant, OR 1.21 (95% CI 1.00-1.47; p=0.055); however, following adjustment for sedentarism and BMI ˃ 25 this association is significant OR=1.68 (95% 1.05-1.56; p=0.033). Finally, in the crude analysis, the relation between positive HAdV-36 and BMI ˃ 25 had a statistically significant odds for obesity OR 1.42 (95% 1.15-1.76; p=0.016), adjusted for BMI ˃ 25 OR=3.37 (95% 1.11-12.51; p=0.015), following further adjusted for low weight and HAdV-36 OR: −1.21, -1.94—0.48; P = 0.001).

Table 8 was add to the manuscript with the linear regression analysis.

Observation: Discussion & Interpretation of Results: Some results are overstated without sufficient supporting evidence. Avoid speculation beyond what the data can substantiate. Findings should be more explicitly compared with previous studies. Address discrepancies and highlight novel contributions. Provide a clearer explanation of the clinical or biological implications of the observed associations.

Answer: The discussion was adjusted by comparing the results of our study with those previously reported. As well as information without sufficient support. The new contributions were highlighted, the limit of founding, as follows:

Discussion

For firs time was described in Mexican population the HAdV-36 seroprevalence and we found a high rate of prevalence positive to HAdV-36 in a sample of Mexican adults between 18 and 28 years old. These prevalence rates were 15% in underweight people (this data was novelty), 18% for normal-weight individuals, and 31% for obese subjects indicated that obese people had higher HAdV-36 seroprevalence, compared with people of normal weight. This data contrasts with the much lower findings than previously described in other countries, for example, the 2015 report Turkish participants a frequency of 17.5% of adults with obesity and 4% without obesity [5]. In our study, 15% of underweight subjects tested positive for HAdV-36 (no obesity group). In Denmark, a very low prevalence of HAdV-36 infection was found in the obese and normal-weight adult populations (5.7% vs. 10.0%) [6]. In Belgian and Netherland individuals there are not a significant association between HAdV-36, and obesity were found, and only 5.5% HAdV-36 positive participants were associated with obesity [7]. In the present study, we demonstrated that the prevalence rate of HAdV-36 in the Mexican population is 28%, these high prevalence which could explain part the etiology of obesity in Mexico, a country with the second-highest rate of obesity worldwide [10], according with the seroprevalences reported in students from University of Wisconsin and the Bowen Center, Naples, Florida from the United States (11% in nonobese vs 30% in obese subjects) [8], these prevalence similarities could be by the Hispanic or Latin participants in the sample. Another study with higher HAdV-36 seroprevalence was in Chilean population with 58% in obese group [21]. The obesity prevalence is higher in Hispanic or Latin people compared with the general population, by the summary of risk factors as genetic, for example, single‐nucleotide polymorphisms have been associated with obesity, high triglyceride levels, and insulin resistance in Mexican American people [25]. The environment factor as diets tends to be high in carbohydrates in Hispanic or Latin population [9], and the infection agents as HAdV-36.

In this study, we did not find significant changes in glucose and insulin levels in HAdV-36 seroprevalence-positive subjects compared to negative subjects. This contrasts with the findings of a study of obese Chilean subjects with positive HAdV-36 seroprevalence, which was associated with insulin resistance and low glucose and insulin levels [21]. Seroprevalence of HAdV-36 has even been associated as a risk factor for type 2 diabetes mellitus [16]. In vitro studies have shown that HAdV-36 infection induces glucose transport in fat cells [14]. In this study, it was a surprise that no significant differences were found in triglyceride levels in both groups with HAdV-36-positive and HAdV-36-negative seroprevalence. Because in children from Chilpancingo, Guerrero, Mexico showed a decrease in HDL levels, compared with an HAdV-36 negative group [26]. Interestingly, in our study we did find that HDL-C and LDL-C levels decreased only in subjects with low-weight BMI and positive HAdV-36 seroprevalence. These data in accord with the lower concentrations of triglycerides and VLDL cholesterol in lean subjects [21] and data from hamsters infected with HAdV-36, both lipoproteins had a negative association with seroprevalence for HAdV-36, and coincides with a change in the lipid profile [27]. In chickens, monkeys and mice, infection with HAdV-36 the mechanism involves adipogenesis, decrease lipolysis, and increase lipogenesis [11, 12].

Obesity is associated with inflammatory alterations, in the present study, subjects with positive HAdV-36 seroprevalence, normal weight, overweight, and obesity had high levels of IL-6, while subjects with low weight the IL-6 levels were not significant. These data are consistent with high IL-6 levels in children positive for HAdV-36 [28, 29], but contrasting in HAdV-36 infected Wistar rats there were no significant changes in IL-6 and TNF-α [30].

The imbalance between oxidative stress and antioxidant could result from excessive fat accumulation, although oxidative stress leads to weight gain by stimulating white adipose tissue deposition and altering food intake [24]. In viral infections, the accumulation of reactive oxygen species causes oxidative stress, which can lead to depletion of the host's antioxidant defenses; this oxidizing environment is used by viruses to replicate, as reported for the H5N1 influenza virus [23]. However, it is unknown if this mechanism of action occupies HAdV-36 for its replication. In the present study, novelty results were found in the subjects with positive seroprevalence for HAdV-36 there were higher levels of oxidative stress, elevated values ​​of cortisol and low levels of catalase activity in overweight and obese subjects with positive seroprevalence for HAdV-36. In addition, subjects with low weight and normal weight who were positive for HAdV-36 presented high levels of glutathione and SOD activity. In the literature there is no evidence about HAdV-36 and oxidative stress, elevated values ​​of cortisol and low levels of catalase activity in overweight and obese subjects with positive seroprevalence for HAdV-36. In addition, subjects with low weight and normal weight who were positive for HAdV-36 presented high levels of glutathione and SOD activity. In the literature no evidence about HAdV-36 and oxidative stress/antioxidant balance was found; however, the imbalance between oxidative stress and antioxidant status results from excessive fat accumulation and oxidative stress leads to weight gain by stimulating white adipose tissue deposition and altering food intake[24], therefore it is possible that HAdV-36 contributes to this established vicious circle.

The present study may limit our findings to participants from rural areas, this was a cross-sectional study, and thus a causality between BMI and HAdV-36 cannot be established and does not include a large random sample. It is proposed prospective studies that enrolling Mexican population to confirm a causal relationship between these variables. In addition to this study design may have influenced the relation of the presence of altered BMI with HAdV-36 with dietary patterns and not influenced by demographic variables related to a BMI increase. However, for first time it is know the seroprevalence of HAdV-36 in a sample of adult apparently healthy Mexican students. In addition, HAdV-36 seroprevalence had relation with altered lipid profiles, inflammation and imbalance between antioxidant and oxidative stress status, this one a novelty data, these changes independently of BMI. This study provides the necessary evidence to initiate new research to understand the mechanism associated between HAdV-36 and oxidative stress and antioxidant status and consider it in future health interventions or prospective studies with Hispanic or Latin subjects.

Conclusion & Future Directions: The Conclusion should better summarize key findings and emphasize their broader implications. Future research directions should be expanded, particularly on mechanistic insights into HAdV-36’s role in obesity and metabolic changes.

Answer: The conclusions of the study and future directions were corrected and remained in the manuscript as follows:

  1. Conclusion

The seroprevalence of HAdV-36 in adult Mexican population for the first time is reported and was higher, and is present in all four BMI categories, including underweight. The positive seroprevalence for HAdV-36 had relation with lipid and inflammation alterations. Finally, for the first time it is reported that there is a relation oxidative stress and antioxidant imbalance against HAdV-36 seroprevalence.

  1. Recommendations

With these results, we can emphasize the need to carry out continuous research to develop comprehensive intervention programs for the prevention and treatment of overweight and obesity in Hispanic or Latin population, which may include appropriate nutrition strategies (antioxidants), physical exercise, and antiviral therapy. Both in vivo and in vitro studies are required to understand the HAdV-36 action mechanism on the balance of oxidative stress and antioxidant status, since HAdv-36 could be using oxidative stress in the deposition of white adipose tissue.

Language & Formatting Issues: The manuscript contains grammatical and typographical errors that require careful proofreading. Ensure that all references are consistently formatted and correctly cited.

Answer: Grammatical errors were corrected, and careful proofreading was performed. However, if it is accepted and errors continue, it will be sent to MDPI's English correction service. In the same way, references were corrected and those that are not relevant were eliminated.

Regards

PhD. Omar Elind Arroyo Helguera

Round 2

Reviewer 1 Report

Comments and Suggestions for Authors

The article has been improved according to my suggestions, and I have nom other doubts that it can be accepted in the current version.

Reviewer 2 Report

Comments and Suggestions for Authors

The revised version is now suitable for publication, as the authors have made the necessary revisions based on my comments.